# OpenReview forum: "In-the-Flow Agentic System Optimization for Effective Planning and Tool Use"
_ICLR.cc/2026/Conference — ICLR 2026 Oral_

### Official Review · Reviewer_1fZQ · 2025-10-28

**Soundness:** 3
**Presentation:** 1
**Contribution:** 2
**Rating:** 6
**Confidence:** 2

**Summary:**

This paper introduces AGENTFLOW, a trainable agentic system that enables multi-turn reasoning and tool use by optimizing its planner in the flow of execution. Traditional tool-augmented LLMs train a single, monolithic policy that struggles with long-horizon reasoning. AGENTFLOW instead decomposes reasoning into four interacting modules that coordinate through a shared evolving memory.
The key technical contribution is Flow-based Group Refined Policy Optimization (Flow-GRPO), an on-policy RL algorithm that converts sparse, long-horizon reinforcement learning into tractable per-turn updates by broadcasting a single trajectory-level reward to all turns and using group-normalized advantages for stability. This design allows for end-to-end optimization of the planner within the live agentic loop.

**Strengths:**

- The paper addresses the sparse reward problem in LLM-based reasoning and enables the training of modular, multi-tool systems in dynamic environments

- strong empirical results

**Weaknesses:**

- representation should be improved; it is super hard to follow

- The reliance on LLM-as-a-Judge rewards risks evaluation leakage. I also find the direct distribution of the global reward across turns unconvincing.

- Although the results are mostly qualitative and good, the paper does not report the computational cost, which raises concerns about efficiency.

**Questions:**

First of all, I would like to thank the authors for their work.
I would like to ask how reward hacking is avoided when global rewards are distributed directly to each turn? In such cases, even undesirable behaviors within a successful trajectory could be reinforced, potentially leading to unnecessarily long or inefficient trajectories. Additionally, could the authors comment on the computational cost of the training process?

---

> ### Author Response · Authors · 2025-11-20
> **Appreciation and Summary of Revisions and New Experiments (Part 1/4)**
>
> We sincerely thank the reviewer for the thoughtful, constructive, and encouraging feedback, and for recognizing both the paper’s contributions and its strong empirical results. Your comments on presentation, reward design, and efficiency directly guided a set of substantial revisions that we believe significantly strengthen the clarity, rigor, and practical impact of the work.
>
> In response, we have made **revisions to [the paper pdf (click here)](https://openreview.net/pdf?id=Mf5AleTUVK)** and conducted **new analyses and experiments** to address all of your insightful comments:
>
> * **W1 (Improve representation):** We improved presentation with a new teaser figure, a reorganized comparison figure in the Preliminary section, a new case-study figure, and clarity/formatting updates to main-text and appendix figures.
>
> * **W2a (Reliance on LLM-as-a-Judge):** We clarified that the LLM-as-a-judge is **final-only and binary**, and added a rule-based “Math Verify and Match” variant, explaining why we still prefer an LLM judge for robust, multi-domain evaluation.
>
> * **W2b (Global reward broadcasting):** We added ablations comparing broadcasted rewards to backward-/forward-discounted and process rewards (3B and 7B). Broadcasting the final outcome consistently achieves the best average performance and stability.
>
> * **W3/Q2 (Computational cost and efficiency):** We now report a detailed cost breakdown for AgentFlow-7B (GPU compute, search API, frozen-module inference, and judge costs) and emphasize that only the planner is trained, making the approach computationally efficient.
>
> * **Q1 (Reward hacking under global rewards):** We explain why our binary, final-only reward limits reward hacking and add robustness experiments with random tool failures, showing that Flow-GRPO improves both accuracy and fault tolerance with smooth performance degradation under noise.
>
> Detailed responses and results for each point are provided below. **We are grateful for your insightful comments and believe these revisions and updates substantially strengthen the paper, and we warmly welcome any further feedback or suggestions you may have.**
>
> ---
>
> ## W1: Improve the representation
> Thank you for the suggestion to improve the paper’s presentation. We have substantially revised the manuscript to enhance clarity and readability. In the main paper, we made the following updates:
>
> - **New teaser figure** summarizing the main results and highlighting Flow-GRPO’s improvements across ten benchmarks *(Figure 1, lines 031-048)*.
> - **Relocated comparison figure** (“Two paradigms of LLMs with tool use”) to the *Preliminary* section, making the exposition more self-contained and easier to follow *(Figure 3, lines 162-176)*.
> - **New case-study figure** illustrating how Flow-GRPO enables AgentFlow to recover from early mistakes and discover improved solution pathways *(Figure 5, lines 378-395)*.
> - **Minor corrections and clarity enhancements** to Figures 2 and 6.
>
> We also improved presentation quality in the Appendix:
>
> - **Updated Figures 11 and 12** for clearer visualization *(lines 1228-1241)*.
> - **Refined badge cards** in Appendix E.1 (instruction templates) and Appendix E.2 (tool metadata).
> - **Enhanced case studies** in Appendix F with clearer intermediate reasoning traces and improved formatting.
>
> Beyond the manuscript, we will provide a **video tutorial**, **live demo**, and **interactive wiki-style documentation** to make the workflow and system design more accessible. These supplementary resources will be released with the final version (after anonymity is lifted) to further support reproducibility.

---

> ### Author Response · Authors · 2025-11-20
> **Appreciation and Summary of Revisions and New Experiments (Part 2/4)**
>
> We thank the reviewer for raising this point. To directly test whether broadcasting a single final-outcome reward might reinforce undesirable intermediate behaviors, we compared our scheme against three alternative reward designs:
>
> 1. **Backward-discounted reward** (final step = 1.0; earlier steps multiplied by a discount factor $k=0.9$).
> 2. **Forward-discounted reward** (first step = 1.0; later steps multiplied by $k=0.9$).
> 3. **Process reward model**, which scores each intermediate step along four dimensions, sub-goal quality, tool-call correctness, consistency with prior steps, and contribution toward the final answer, each in $[-1, 1]$.
>
> We report both 3B- and 7B-backbone results below.
>
> **3B backbone**
>
> ||Bamboogle|2Wiki|HotpotQA|Musique|GAIA|AIME24|AMC23|GameOf24|GPQA|MedQA|Avg
> |-|-|-|-|-|-|-|-|-|-|-|-
> |**broadcasting reward (ours)**|68.8|72.3|66.0|20.2|29.1|20.0|75.0|53.0|38.0|81.0|**52.3**
> |discount reward^1|71.2|69.0|67.0|21.0|20.5|26.7|52.5|14.0|34.0|79.0|**45.5**
> |reversed discount^2|69.6|59.6|68.7|15.0|20.0|16.7|45.0|6.0|68.7|82.5|**45.2**
> |process reward^3|56.8|56.6|61.0|19.0|23.2|16.7|50.0|26.0|36.0|77.0|**42.2**
>
> **7B backbone**
> ||Bamboogle|2Wiki|HotpotQA|Musique|GAIA|AIME24|AMC23|GameOf24|GPQA|MedQA|Avg
> |-|-|-|-|-|-|-|-|-|-|-|-
> |**broadcasting reward (ours)**|69.6|77.2|57.0|25.3|33.1|40.0|61.5|53.0|47.0|80.0|**54.4**
> |discount reward^1|67.2|63.6|70.0|23.0|27.6|23.3|57.5|35.0|43.0|82.0|**49.2**
> |reversed discount^2|60.0|57.0|67.0|18.4|26.2|6.7|52.5|13.0|29.0|79.8|**41.0**
> |process reward^3|67.2|72.0|68.0|19.0|28.6|43.3|65.0|35.0|41.0|78.0|**51.7**|
>
>  **Key findings.**
>  Across both model sizes, **broadcasting the global reward achieves the best average performance** (52.3% for 3B and 54.4% for 7B), outperforming backward-discounted (45.5% / 49.2%), forward-discounted (45.2% / 41.0%), and process-based rewards (42.2% / 51.7%). The broadcast scheme also yields strong and *stable* gains across diverse tasks (e.g., Bamboogle, 2Wiki, GAIA, GameOf24), rather than overfitting to a particular domain.
>
> **Why broadcasting works best.**
>
> Empirically, we find that broadcasting a single final-outcome reward leads to more stable optimization and better reasoning quality for three main reasons:
>
> **1. Low-variance, outcome-aligned signal.**
>
> Broadcasting gives every turn the same clear signal tied directly to final correctness. This reduces variance in the multi-turn setting and avoids the instability we observed when step-level rewards were noisy or inconsistent.
>
> **2. Fine-grained credit assignment is challenging with generic step rewards.**
>
> Discounted and process rewards require **accurate automatic evaluation of intermediate reasoning quality**. In our heterogeneous, multi-domain setting, **step-level evaluation is intrinsically harder than final-outcome judging**: the judge must score partially formed thoughts, sub-goals, and tool calls that are not yet verifiable, which leads to higher-variance signals and less reliable credit assignment. Empirically, as results shown above, **these step-level rewards produced noisier learning dynamics** and worse performance than using a single, verifiable final-outcome reward that directly reflects task success.
>
> We note that, in principle, more task-specific or structure-aware process rewards (e.g., programmatic checkers or domain-specific verifiers for a **single** benchmark) could further improve credit assignment. However, systematically designing and tuning such rewards across our ten heterogeneous benchmarks is beyond the scope of this work; our focus here is to compare **generic, broadly applicable reward** schemes under a unified framework.
>
> **3. Reduced exposure to judge bias and reward hacking.**
>
> More granular rewards amplify biases of the judge model and can inadvertently reward superficially “clean” but unhelpful steps, while penalizing unconventional yet effective reasoning. By using only the final outcome, the broadcast scheme minimizes these biases and limits the opportunities to exploit brittle step-wise heuristics.
>
> **Conclusion:** While more structured reward designs may appear preferable in principle, our ablations show that **the simplest scheme, broadcasting the final reward to all turns, is the most stable, robust, and empirically effective** for long-horizon agentic optimization. This is why we adopt it as the primary reward mechanism in the main paper.

---

> ### Author Response · Authors · 2025-11-20
> **Appreciation and Summary of Revisions and New Experiments (Part 3/4)**
>
> ## W2b: Reliance on LLM-as-a-Judge rewards
> We appreciate the reviewer’s thoughtful comment. In our setup, the LLM-as-a-Judge plays a **strictly limited, binary, and final-only** role: it checks whether the model’s **final answer matches the ground-truth label in the training corpus**, and nothing more. It does **not** evaluate intermediate reasoning, provide partial credit, or inject external knowledge. This design prevents evaluation leakage and minimizes reward hacking risks.
>
> Below, we clarify (1) why this design makes reward hacking and leakage unlikely, (2) how a purely rule-based judge performs in practice, and (3) why we still prefer an LLM-based judge for scalability across domains.
>
> **1. Final-only, binary rewards limit reward hacking and leakage**
>
> Our reward function is strictly binary:
>
> - **1** if the final generated answer matches the ground truth,
> - **0** otherwise.
>
> There is **no process reward, no separate reward model, no shaping heuristics, and no intermediate feedback**. The judge only sees the final answer and the dataset label, and returns a 0/1 signal. This is equivalent in spirit to automated exact-match or numeric-equality scoring, and greatly limits opportunities for reward hacking: undesirable steps within a trajectory are not directly rewarded unless they still lead to a correct final answer. Crucially, the judge does **not** provide gradients on partial reasoning or style, which are typical channels for reward hacking in more expressive reward models.
>
> **2. Rule-based verification is possible but brittle in multi-domain RL**
>
> To directly address the concern, we implemented a **purely rule-based variant (“Math Verify and Match”)** that replaces the LLM-as-a-Judge with:
>
> - numerical equivalence checks for math tasks, and
> - exact string match for non-math tasks.
>
> The results for 7B-backbone AgentFlow are:
>
> ||Bamboogle|2Wiki|HotpotQA|Musique|GAIA|AIME24|AMC23|Avg
> |-|-|-|-|-|-|-|-|-
> |LLM-as-a-Judge|69.6|77.2|57.0|25.3|33.1|40.0|61.5|**52.0**
> |Math Verify and Match|67.2|79.0|55.0|21.0|26.0|36.7|55.0|**48.6 (-3.4)**
>
> While rule-based verification slightly outperforms in a few cases (e.g., 2Wiki), it degrades substantially on others (e.g., Musique, GAIA, GameOf24), with **an average accuracy drop of 3.4%** across these benchmarks. This pattern reflects fundamental constraints of rule-based evaluation.
>
> **3. Why we use an LLM-based judge in practice**
>
> We ultimately adopt an LLM-as-a-Judge for **practical scalability**, not as a source of additional information:
>
> - It handles heterogeneous answer formats (LaTeX, natural language, symbolic expressions) more robustly than rigid parsers.
> - It reduces the need for many task-specific evaluators (e.g., datasets like GameOf24 that require both correctness and structural constraints).
> - It mitigates false negatives due to minor formatting variations, which become more common as RL encourages exploratory outputs.
>
> Importantly, the judge is only used to turn **known ground-truth labels** into consistent binary rewards across diverse tasks. It does not supply missing knowledge or intermediate signals, and we show that a rule-based alternative behaves similarly where it is reliable, but is significantly more brittle across domains.
>
> In summary, our reliance on an LLM-as-a-Judge is **narrow (final-only, binary), replaceable (via rule-based variants), and motivated by scalability**, which together make evaluation leakage and reward hacking unlikely in our setting.
>
> ---
> ## W3/Q2: Report of computational cost and efficiency
> We thank the reviewer for prompting a discussion of computational cost and training efficiency.
>
> We now include a detailed cost breakdown of a typical training run for **AgentFlow-7B** with Flow-GRPO as below:
>
> **1. GPU Computation (Training)**
>
> - Because Flow-GRPO converges quickly and we only update the planner module, training is rapid and highly efficient.
> - A full run consists of **68 RL steps**.
> - Using a cloud rate of \$5.33 per 8 GPU-hours, the cost is **$63.4**.
>
> **2. External Tooling (Search)**
>
> - The primary external cost arises from Google Search API, which consumed 9.98K queries, costing **$72.0**.
>
> **3. Frozen Module Inference**
> - The non-trainable modules (Executor, Verifier, Generator) processed 121M input tokens and 7.7M output tokens.
> - Using a commercial API (e.g., DashScope), this costs **$20.50**.
> - However, our codebase supports running these modules locally, which reduces this specific marginal cost to **zero**.
>
> **4. Reward Evaluation**
>
> - LLM-as-a-judge costs for reward calculation were negligible, totaling roughly **$4.0**.
>
> **Total Cost Summary**
>
> |Category|Cost
> |-|-
> |GPU Compute|$63.4
> |External Tooling|$72.0
> |Module Inference (non-local)|$20.5
> |Reward Eval|$4.0
> |**Total (cloud setting)**|**≈$160**
> |**Total (with local modules)**|**≈$140**

---

> ### Author Response · Authors · 2025-11-20
> **Appreciation and Summary of Revisions and New Experiments (Part 4/4)**
>
> ## Q1: How reward hacking is avoided when global rewards are distributed directly to each turn? Undesirable behaviors could be reinforced.
> We thank the reviewer for this important question. We address it by (i) clarifying our reward design and (ii) providing new robustness experiments that probe whether the planner learns degenerate behaviors.
>
> **1. Reward design: why reward hacking is limited in our setting**
>
> **(a) The judge only verifies the *final* answer, not the process.**
>
> Our LLM-as-a-judge is used purely as a *result verifier*: it checks whether the final prediction matches the ground-truth answer (semantic / numeric / option-level equivalence) and returns a binary reward in ${0,1}$. It *does not* grade intermediate steps. As a result:
>
> * There is no way to farm partial credit for looking systematic or producing long chains of thought.
> * To obtain high reward, the planner must actually produce a correct final answer; incorrect trajectories receive reward 0.
>
> **(b) Broadcasting is the most conservative, outcome-aligned choice.**
>
> Because correctness of intermediate steps is highly task- and judge-dependent, step-wise rewards are much more subjective and noisy. Broadcasting the final outcome to all turns is conservative:
>
> * It uses the most objective signal available (final correctness), avoiding hand-crafted heuristics for intermediate goodness.
> * Combined with KL regularization to a reference policy and group-normalized advantages, it encourages policies that are both outcome-aligned and close to the reference model, which empirically discourages unnecessarily long or unnatural trajectories.
>
> As also shown in our training-dynamics analysis (Fig.8, Section 4.5), response lengths **stabilize** during Flow-GRPO training rather than exploding, suggesting that the model is not exploiting redundant steps to “hack” the reward.
>
> **2. New robustness experiments: behavior under random tool failures**
>
> To further test whether broadcasting encourages brittle or exploitative behaviors, we evaluated \model under **random tool failures** during both training and inference. In addition to natural tool errors, we injected independent failures with probability $p \in {0.1, 0.2, 0.4, 0.6}$; on failure, the tool returned a standardized error message (“Tool execution failed due to random error (dropout simulation)”). This setup emulates dynamic or adversarial tool environments where naive reward-hacking strategies (e.g., blindly overusing tools or relying on fragile patterns) should break down.
>
> **Before Flow-GRPO tuning (7B)**
>
> ||Bamboogle|2Wiki|HotpotQA|Musique|GAIA|AIME24|AMC23|GameOf24|GPQA|MedQA|Avg|Gain
> |-|-|-|-|-|-|-|-|-|-|-|-|-
> |No tool error|58.4|60.0|51.3|19.2|17.2|16.7|47.4|31.0|37.0|76.0|**41.4**|/
> |$p=0.1$ tool error|59.2|61.6|60.6|20.0|22.8|6.7|50.0|16.0|30.0|67.0|**39.4**|**−2.0**
> |$p=0.2$ tool error|61.6|58.6|63.0|22.0|22.1|16.7|55.0|12.0|31.0|66.0|**40.8**|**−0.6**
> |$p=0.4$ tool error|59.7|60.2|61.6|18.0|19.7|20.0|47.5|17.0|28.0|66.7|**39.8**|**−1.6**
> |$p=0.6$ tool error|57.6|55.6|57.6|17.0|14.2|10.0|42.5|14.0|22.0|63.0|**35.3**|**−6.1**
>
> **After Flow-GRPO tuning (7B)**
>
> ||Bamboogle|2Wiki|HotpotQA|Musique|GAIA|AIME24|AMC23|GameOf24|GPQA|MedQA|Avg|Gain
> |-|-|-|-|-|-|-|-|-|-|-|-|-
> |No tool error|69.6|77.2|57.0|25.3|33.1|40.0|61.5|53.0|47.0|80.0|**54.4**|**+13.0**
> |$p=0.1$ tool error|71.2|69.7|71.0|20.0|34.1|36.7|70.0|38.0|41.0|65.0|**51.7**|**+10.3**
> |$p=0.2$ tool error|67.2|64.0|69.0|25.0|31.5|43.3|70.0|36.0|41.0|65.0|**51.2**|**+9.8**
> |$p=0.4$ tool error|64.8|68.0|67.0|21.0|26.8|26.7|70.0|35.0|36.0|63.0|**47.8**|**+6.4**
> |$p=0.6$ tool error|63.5|63.0|62.0|18.0|21.4|10.0|42.5|16.0|34.0|68.0|**39.8**|**−1.6**
>
> **Findings**
>
> * **F1: Static agentic systems are fragile.** Before tuning, performance drops markedly under high tool failure rates (41.4%→35.3% at $p=0.6$).
>
> * **F2: Flow-GRPO improves robustness, not just raw accuracy.** After training, AgentFlow maintains strong performance under mild-moderate failures ($p=0.1,0.2,0.4$: 51.7%, 51.2%, 47.8% vs. 41.4% baseline).
>
> * **F3: Flow-GRPO remains stable even under extreme failures.**
> Under heavy tool noise ($p=0.6$), pre-tuning AgentFlow average accuracy drops by 6.1% (41.4%→35.3%), whereas tuned AgentFlow stays within 1.6% of the clean pre-tuning score (39.8% vs. 41.4%) and still significantly outperforms the noisy pre-tuning counterpart (35.3%→39.8%, +4.5%).
>
> * **F4: Degradation is smooth and controlled.** As tool noise increases, the tuned planner degrades gracefully rather than collapsing, and qualitative analysis (Fig. 5)  shows it learns to re-plan after failures, switch tools, and repair memory.
>
> **Conclusion:** (i) The use of a binary final-outcome verifier, (ii) the conservative choice of reward broadcasting with KL regularization, and (iii) the robustness under substantial tool noise together suggest that Flow-GRPO does not encourage reward hacking, but instead learns policies that are both accurate and resilient in long-horizon settings.

---

> ### Author Response · Authors · 2025-11-26
> **Appreciation and Follow-Up Discussion for Reviewer 1fZQ**
>
> Dear Reviewer 1fZQ,
>
> Thank you again for your thoughtful and encouraging review, and for your positive overall assessment of our work. We truly appreciate the time you spent reading the paper, engaging with the ideas, and considering our initial response.
>
> ## Initial response and revisions (2025-11-19)
>
> **In [our first response](https://openreview.net/forum?id=Mf5AleTUVK&noteId=moUqNIYCO7), we revised [the paper](https://openreview.net/pdf?id=Mf5AleTUVK)** and added analyses and experiments addressing your comments on:
>
> * representation and clarity of the framework and training procedure,
> * the role and robustness of the LLM-as-a-judge reward,
> * computational cost and efficiency, and
> * generalization and scaling behavior.
>
> Since then, we have conducted **additional experiments** to further strengthen the empirical and methodological support for **AgentFlow** and **Flow-GRPO**. We summarize these new results below.
>
> ## 1. Inference efficiency and cost reduction (Q1)
>
> We extend cost analysis with **latency and tool-usage statistics** before and after RL tuning:
>
> |Metric|Bamboogle|2Wiki|HotpotQA|Musique|GAIA
> |-|-|-|-|-|-
> |**Before RL tuning**
> |Turns avg|3.1|3.8|4.0|5.6|4.4
> |Time avg (s)|127.3|170.9|98.6|184.6|265.7
> |Google Search calls|0.85|1.06|1.10|2.14|0.79
> |LLM calls|0.39|1.07|0.79|2.13|1.90
> |**After RL tuning**
> |Turns avg|2.1|4.4|3.9|5.4|5.4
> |Time avg (s)|44.1|106.2|85.6|162.7|172.5
> |Google Search calls|1.50|3.16|1.21|1.99|3.34
> |LLM calls|0.02|0.18|0.89|1.81|0.64
> |**Improvement**
> |Time reduction|-65.4%|-37.9%|-13.2%|-11.9%|-35.1%
> |LLM calls reduction|-94.9%|-83.2%|+12.7%|-15.0%|-66.3%
>
> These results suggest that Flow-GRPO not only **improves accuracy**, but also yields **better cost-performance trade-offs** by encouraging more efficient tool and LLM usage.
>
> ## 2. Additional evidence for stability of rewards: robustness with train-time simulated tool errors (Q1)
>
> In our earlier response, we reported robustness under **clean training** with **test-time error injection** for Qwen2.5-7B:
>
> |Test time error|Avg Acc (before tuning)|Avg Acc (after tuning)|Gain
> |-|-|-|-
> |p=0.0|41.4|54.4|**+13.0**
> |p=0.1|39.4|51.7|**+12.3**
> |p=0.2|40.8|51.2|**+10.4**
> |p=0.4|39.8|47.8|**+8.0**
> |p=0.6|35.3|39.8|**+4.5**
>
> To more directly test stability of the reward design and learning dynamics in **dynamic environments**, we now introduce **simulated tool failures during training** as well. Each tool call independently returns an error with probability $p \in {0.1, 0.2, 0.4, 0.6}$, mimicking real-world settings where tools may fail, time out, or return malformed outputs.
>
> |Test time error|Avg Acc (before tuning)|Avg Acc (after tuning)|Gain
> |-|-|-|-
> |p=0.0|41.4|54.4|**+13.0**
> |p=0.1|39.4|51.0|**+11.6**
> |p=0.2|40.8|52.9|**+12.1**
> |p=0.4|39.8|52.2|**+12.4**
> |p=0.6|35.3|47.1|**+11.8**
>
> When **both training and evaluation environments are noisy**, AgentFlow still maintains **large gains over the untuned system**. Under extreme noise ($p = 0.6$):
>
> * tuned AgentFlow achieves **47.1%** average accuracy versus **35.3%** for the untuned system (**+11.8%**), and
> * it even exceeds the untuned system under clean evaluation (41.4%, **+5.7%**).
>
> These results provide further evidence that training with a broadcasted final-outcome reward and Flow-GRPO leads to **stable, resilient policies** in environments where tool reliability is imperfect.
>
> ## 3. Additional evidence for generalization: training on fewer tools, evaluating on all tools (W3/Q2)
>
> We also study a stricter **tool generalization** setting, where the planner is **trained with only 2 tools** (`Python Coder`, `Google Search`) but **evaluated with the full 5-tool set** (`Base Generator`, `Python Coder`, `Wikipedia Search`, `Google Search`, `Web Search`):
>
> |Tuning|Test  time|Bamboogle|2Wiki|HotpotQA|Musique|GAIA|AIME24|AMC23|GameOf24|GPQA|MedQA|Avg
> |-|-|-|-|-|-|-|-|-|-|-|-|-
> |Before tuning|5 tools|58.4|60.0|58.3|19.2|17.2|16.7|47.4|31.0|37.0|76.0|42.1
> |Tuning with 2 tools|5 tools|78.4|71.0|70.0|23.5|32.3|26.7|52.5|44.0|29.0|81.8|**50.9**|
> |**Gain**|/|+20.0|+11.0|+11.7|+4.3|+15.1|+10.0|+5.1|+13.0|-8.0|+5.8|**+8.8**
>
> Even though the planner only uses 2 tools during RL, it achieves an **average gain of +8.8%** when evaluated with the full toolset. This suggests that Flow-GRPO encourages learning **general planning and tool-selection strategies** that transfer to richer, partially unseen tool inventories, rather than overfitting to a specific configuration.
>
> ## Kind request for your feedback
>
> We hope these additional experiments, together with the clarifications and results in our initial response, further strengthen the empirical and methodological evidence for **AgentFlow** and **Flow-GRPO**.
>
> We would be very grateful for any **further feedback or suggestions** you may have, which could help us continue improving the paper and making the framework more useful to the community.
>
> Thank you again for your time, thoughtful evaluation, and supportive review.
>
> Sincerely,
>
> The Authors

---

### Official Review · Reviewer_YoHg · 2025-10-28

**Soundness:** 3
**Presentation:** 3
**Contribution:** 3
**Rating:** 8
**Confidence:** 3

**Summary:**

This work introduces AgentFlow, a trainable, in-the-flow agentic framework that coordinates four modules through an evolving memory, optimizing the planner on-policy within the multi-turn reasoning loop. It further proposes Flow-GRPO, a novel reinforcement learning algorithm that converts long-horizon, sparse-reward multi-turn optimization into tractable single-turn updates by broadcasting a trajectory-level reward to all turns. Experiments across benchmarks demonstrate substantial gains over state-of-the-art models.

**Strengths:**

1. The idea of in-the-flow reinforcement learning for agentic systems is both useful and interesting.
2. The proposed Flow-GRPO provides a stable, elegant formulation for long-horizon credit assignment.
3. The authors conduct comprehensive evaluation across multiple domains, outperforming competitive baselines.

**Weaknesses:**

1. The authors conduct comprehensive experiments. It's better to add discussion on computational cost and training stability.
2. The proposed method performs good on text-based tasks. How about the results on dynamic environments or multi-modal settings?
3. Minor: It's better to add some task description presented in appendix in Figure 1 or 2 to make the solution more clear.

**Questions:**

It's a well-written and comprehensive paper. Some potential improvements please refer to the Weakness section.

---

> ### Author Response · Authors · 2025-11-20
> **Appreciation and Summary of Revisions and New Experiments (Part 1/3)**
>
> We sincerely thank the reviewer for the thoughtful, constructive, and encouraging feedback. We greatly appreciate your recognition of the paper’s technical contributions and comprehensive evaluation, and we have further strengthened the work along all of the dimensions you highlighted.
>
> In response, we have made **substantial revisions to [the paper pdf (click here)](https://openreview.net/pdf?id=Mf5AleTUVK)** and conducted **new analyses and experiments** to address all of your valuable points:
>
> * **W1a:** Added a detailed discussion of **computational cost**, including a full cost breakdown of GPU compute, tool usage, frozen-module inference, and reward evaluation.
> * **W1b:** Added an expanded analysis of **training stability**, together with **new controlled experiments** across model scales (1.5B-8B) showing stable and consistent Flow-GRPO optimization.
> * **W2b:** Added **multimodal experiments** (MathVista, VQA v2, CLEVR-Math) using visual tools, showing that Flow-GRPO generalizes naturally to multimodal agentic reasoning.
> * **W3:** Improved clarity in Figures 1-3 and Appendix figures, added task descriptions, and reorganized visuals for easier comprehension.
>
> Detailed responses and results are provided below. We believe these revisions significantly reinforce both the technical message and practical impact of the paper, and we sincerely appreciate your helpful comments. **We also welcome any further feedback or suggestions you may have.**
>
> ---
> ## W1a: Add discussion on computational cost
> We thank the reviewer for prompting a discussion of computational cost and training efficiency. AgentFlow is **highly efficient**; a full training run for AgentFlow-7B costs about $160 using standard cloud rates, or less if resources are hosted locally.
>
> Below is a detailed cost breakdown of a typical training run for **AgentFlow-7B** with Flow-GRPO:
>
> **1. GPU Computation (Training)**
>
> - Because Flow-GRPO converges quickly and we only update the planner module, training is rapid and highly efficient.
> - A full run consists of **68 RL steps**.
> - On 8×A100 GPUs, a full run takes roughly **12 hours** (95.3 GPU-hours).
> - Using a cloud rate of \$5.33 per 8 GPU-hours, the cost is **$63.4**.
>
> **2. External Tooling (Search)**
>
> - The primary external cost arises from Google Search API, which consumed 9.98K queries, costing **$72.0**.
>
> **3. Frozen Module Inference**
> - The non-trainable modules (Executor, Verifier, Generator) processed 121M input tokens and 7.7M output tokens.
> - Using a commercial API (e.g., DashScope), this costs **$20.50**.
> - However, our codebase supports running these modules locally, which reduces this specific marginal cost to **zero**.
>
> **4. Reward Evaluation**
>
> - LLM-as-a-judge costs for reward calculation were negligible, totaling roughly **$4.0**.
>
> **Total Cost Summary**
>
> |Category|Cost
> |-|-
> |GPU Compute|$63.4
> |External Tooling|$72.0
> |Module Inference (non-local)|$20.5
> |Reward Eval|$4.0
> |**Total (cloud setting)**|**≈$160**
> |**Total (with local modules)**|**≈$140**

---

> ### Author Response · Authors · 2025-11-20
> **Appreciation and Summary of Revisions and New Experiments (Part 2/3)**
>
> ## W1b: Add discussion on training stability
> We thank the reviewer for emphasizing the importance of discussing training stability.
>
> In the main submission, we analyze stability in **Section 4.3** (comparing different planner-training strategies) and **Section 4.5** (contrasting Flow-GRPO with prior tool-integrated RL methods such as ToRL).
>
> To further address this point, we conducted **additional experiments** to evaluate the robustness of Flow-GRPO under significantly smaller backbones and across alternative architectures, two settings where long-horizon RL is typically unstable.
>
> Below we summarize these new findings, which consistently show that Flow-GRPO maintains **stable, monotonic optimization** even in challenging regimes.
>
> **1. New Experiments: Stable Optimization on 1.5B Models**
>
> To stress-test stability, we instantiated all four modules using **Qwen2.5-1.5B-Instruct**, a compact backbone where RL fine-tuning often destabilizes due to limited capacity. Despite this, Flow-GRPO trains smoothly and produces **large, consistent gains** across all ten benchmarks:
>
> ||Bamboogle|2Wiki|HotpotQA|Musique|GAIA|AIME24|AMC23|GameOf24|GPQA|MedQA|Avg
> |-|-|-|-|-|-|-|-|-|-|-|-
> |**Before tuning**|7.23|18.0|15.0|5.0|1.6|0.0|12.5|1.0|21.0|53.0|13.4
> |**After tuning**|43.7|56.8|50.0|13.1|15.7|13.3|37.5|27.5|31.0|55.0|**34.4**
> |**Gain**|**+36.4**|**+38.8**|**+35.0**|**+8.1**|**+14.1**|**+13.3**|**+25.0**|**+26.5**|**+10.0**|**+2.0**|**+21.0**
>
> **Analysis**
>
> - Despite its limited capacity, AgentFlow with Flow-GRPO trains smoothly and delivers large, uniform improvements (a massive **+21.0%** average accuracy gain) across all ten benchmarks.
> - Notably, it unlocked capabilities in hard reasoning tasks like AIME24 (0%→13.3%) and GameOf24 (1.0%→27.5%).
> - No instability or collapse is observed, confirming that Flow-GRPO remains stable even for small, RL-sensitive backbones.
>
> **2. New Experiments: Consistent and Stable Gains on Llama 3B and 8B Models**
>
> We further validated stability by applying Flow-GRPO to **Llama 3.2 3B** and **Llama 3.1 8B**, both structurally distinct from Qwen. In both cases, training remains smooth and consistently improves performance across diverse tasks:
>
> **Llama 3.2 3B**
>
> ||Bamboogle|2Wiki|HotpotQA|Musique|GAIA|AIME24|AMC23|GameOf24|GPQA|MedQA|Avg
> |-|-|-|-|-|-|-|-|-|-|-|-
> |**Before tuning**|36.8|30.0|48.0|12.0|7.9|3.3|20.0|2.0|31.0|76.0|26.7|
> |**After tuning**|64.0|60.0|52.0|13.0|21.3|23.3|30.0|41.0|37.0|74.0|**41.6**|
> |**Gain**|**+27.2**|**+30.0**|**+4.0**|**+1.0**|**+13.4**|**+20.0**|**+10.0**|**+39.0**|**+6.0**|**–2.0**|**+14.9**
>
> **Llama 3.1 8B**
>
> ||Bamboogle|2Wiki|HotpotQA|Musique|GAIA|AIME24|AMC23|GameOf24|GPQA|MedQA|Avg
> |-|-|-|-|-|-|-|-|-|-|-|-
> |**Before tuning**|63.2|55.0|61.0|18.0|26.0|16.7|20.0|20.0|30.0|75.0|38.5
> |**After tuning**|67.2|72.0|70.0|18.0|29.1|16.7|50.0|31.0|41.0|88.0|**48.3**|
> |**Gain**|**+4.0**|**+17.0**|**+9.0**|**0.0**|**+3.2**|**0.0**|**+30.0**|**+11.0**|**+11.0**|**+13.0**|**+9.8**
>
> **Analysis**
>
> - **Improvements are smooth, monotonic**, and free of collapse or oscillation.
> - The gains persist across different architectures, parameter sizes, and task types.
> - These results show that Flow-GRPO maintains stable optimization beyond the Qwen series.
>
> **3. Takeaway: Stable and Generalizable RL Optimization**
>
> Across Qwen-1.5B, Llama-3B, and Llama-8B, Flow-GRPO achieves:
>
> - **Stable on-policy training**, even for small models that usually destabilize during RL.
> - **Consistent cross-task gains** on all ten benchmarks.
> - **Architecture-agnostic robustness**, demonstrating that Flow-GRPO is not tailored to a specific model family.
>
> These expanded experiments strengthen the claim that **Flow-GRPO is a stable, scalable RL method** for agentic systems operating under long-horizon and sparse-reward conditions.
>
> ---
> ## W2a: How about the results on multi-modal settings?
> To address the reviewer’s question about multi-modal capabilities, we extended AgentFlow with three additional **visual tools** to enable image-based reasoning:
>
> - `Visual Generator`: answer general multimodal queries
> - `Image Captioner`: generate image captions
> - `OCR Detector`: extract text from images
>
> Only the **planner** was trained; all visual modules remained *frozen*.
>
> ||MathVista|VQA v2|CLEVR-Math|Avg.|
> |-|-|-|-|-|
> |**Before training 7B**|60.0|70.0|64.0|64.7|
> |**After training 7B**|64.0|76.0|73.0|71.0|
> |**Gain**|**+4.0**|**+6.0**|**+9.0**|**+6.3**|
>
> **Analysis**: AgentFlow yields a **+6.3% avg. gain**, which arises from **improved tool orchestration**: the planner learns when and how to invoke visual tools and how to integrate visual signals with symbolic reasoning (e.g., in MathVista and CLEVR-Math). Notably, **no architectural modifications were required**: simply adding visual tools enabled strong cross-modal generalization, underscoring the flexibility of both AgentFlow and Flow-GRPO.

---

> ### Author Response · Authors · 2025-11-20
> **Appreciation and Summary of Revisions and New Experiments (Part 3/3)**
>
> ## W2b: How about the results on dynamic environments?
> We thank the reviewer for highlighting the importance of evaluating robustness under dynamic environments.
>
> To address this, we conducted additional experiments introducing **random tool failure during training and inference**. Beyond each tool’s natural error rate, we injected independent failure events with probability ($p \in {0.1, 0.2, 0.4, 0.6}$). Upon failure, the tool returned a standardized message (“Tool execution failed due to random error (dropout simulation)”), simulating dynamic or adversarial environments where tools intermittently malfunction or return corrupted outputs.
>
> **Before tuning (7B):**
>
> ||Bamboogle|2Wiki|HotpotQA|Musique|GAIA|AIME24|AMC23|GameOf24|GPQA|MedQA|Avg|Gain
> |-|-|-|-|-|-|-|-|-|-|-|-|-
> |No tool error|58.4|60.0|51.3|19.2|17.2|16.7|47.4|31.0|37.0|76.0|**41.4**|/
> |$p=0.1$ tool error|59.2|61.6|60.6|20.0|22.8|6.7|50.0|16.0|30.0|67.0|**39.4**|**−2.0**
> |$p=0.2$ tool error|61.6|58.6|63.0|22.0|22.1|16.7|55.0|12.0|31.0|66.0|**40.8**|**−0.6**
> |$p=0.4$ tool error|59.7|60.2|61.6|18.0|19.7|20.0|47.5|17.0|28.0|66.7|**39.8**|**−1.6**
> |$p=0.6$ tool error|57.6|55.6|57.6|17.0|14.2|10.0|42.5|14.0|22.0|63.0|**35.3**|**−6.1**
>
> **After tuning (7B):**
>
> ||Bamboogle|2Wiki|HotpotQA|Musique|GAIA|AIME24|AMC23|GameOf24|GPQA|MedQA|Avg|Gain
> |-|-|-|-|-|-|-|-|-|-|-|-|-
> |No tool error|69.6|77.2|57.0|25.3|33.1|40.0|61.5|53.0|47.0|80.0|**54.4**|**+10.0**
> |$p=0.1$ tool error|71.2|69.7|71.0|20.0|34.1|36.7|70.0|38.0|41.0|65.0|**51.7**|**+10.3**
> |$p=0.2$ tool error|67.2|64.0|69.0|25.0|31.5|43.3|70.0|36.0|41.0|65.0|**51.2**|**+9.8**
> |$p=0.4$ tool error|64.8|68.0|67.0|21.0|26.8|26.7|70.0|35.0|36.0|63.0|**47.8**|**+6.4**
> |$p=0.6$ tool error|63.5|63.0|62.0|18.0|21.4|10.0|42.5|16.0|34.0|68.0|**39.8**|**−1.6**
>
> **Finding 1: The pre-tuning system is brittle under substantial tool noise.**
>
> Under the 7B backbone **before Flow-GRPO RL tuning**, performance drops rapidly as failure probability increases (Avg: **41.4%→35.3%** at $p=0.6$). This confirms that **static, training-free agentic systems are highly brittle**, where a single faulty tool call can derail the entire reasoning trajectory.
>
> **Finding 2: Our Flow-GRPO RL tuning markedly improves robustness.**
>
> After Flow-GRPO tuning, AgentFlow remains stable across all noise levels:
>
> - Under **mild to moderate failures** ($p = 0.1, 0.2, 0.4$), the tuned system retains **strong performance** (Avg: **51.7%, 51.2%, 47.8%**), consistently outperforming the untuned baseline (41.4%) by **+10.3%, 9.8%, 6.4%**.
> - Even with **extreme noise** ($p = 0.6$), performance remains comparable to the **clean** pre-tuning condition:
>   - Pre-tuning ($p=0.0$): **41.4%**
>   - Post-tuning ($p=0.6$): **39.8%**
>   - Crucially, tuning still provides clear gains over the untuned model under the *same* high-noise condition (Avg: **35.3%→39.8%, +4.5%**).
> - Across all 10 benchmarks and failure probabilities, the tuned AgentFlow **strictly outperforms** the untuned system, showing that Flow-GRPO reliably strengthens fault tolerance.
>
> **Finding 3: Flow-GRPO yields smoother, more adaptive degradation**
>
> While performance naturally declines as tool noise increases, the RL-tuned planner exhibits a **much smoother degradation curve**. Flow-GRPO teaches the planner to:
>
> - re-plan after failed calls,
> - fall back to alternative tools,
> - validate or repair memory state, and
> - avoid cascading errors from a single faulty observation.
>
> These adaptive behaviors are absent in the static, pre-tuned system.
>
> **Conclusion:** These robustness experiments show that Flow-GRPO improves not only standard accuracy but also **resilience in dynamic or adversarial environments** with unreliable tools. This directly supports the reviewer’s suggestion and further demonstrates the benefits of in-the-flow optimization for stable multi-turn reasoning.
>
> ---
> ## W3: “Minor: It's better to add some task description presented in appendix in Figure 1 or Figure 2”
> Thank you very much for this suggestion. We would appreciate a bit more clarification on which specific figures you are referring to, as the Appendix begins with **Figure 8** in our previous submission. Could you kindly let us know which appendix figures you had in mind?
>
> As noted in **Part 1/3**, we have made substantial revisions to the
> **[the paper pdf (click here)](https://openreview.net/pdf?id=Mf5AleTUVK)**, including adding new figures, reorganizing and improving figures, and enhancing clarity across both the main text and Appendix.
>
> We hope these updates address your suggestion, and we would be happy to refine any specific appendix figures you had in mind.

---

> ### Author Response · Authors · 2025-11-26
> **Appreciation and Follow-Up Discussion for Reviewer YoHg**
>
> Dear Reviewer YoHg,
>
> Thank you again for your thoughtful, constructive, and encouraging review, including your high evaluation and recommendation for acceptance. We truly appreciate the time you spent reading both the paper and our initial response.
>
> ## Initial response on 2025/11/19
>
> **In [our first response](https://openreview.net/forum?id=Mf5AleTUVK&noteId=tFPGt4lylL), we revised [the paper](https://openreview.net/pdf?id=Mf5AleTUVK) and added analyses and experiments** addressing your comments on generalization, computational cost, scaling behavior, and verifier robustness.
>
> Since then, we have run additional experiments to further strengthen the empirical and methodological support for **AgentFlow** and **Flow-GRPO**. We summarize new results below.
>
> ## 1. Inference efficiency and cost-performance trade-offs (W1a)
>
> We extend cost analysis with **latency and tool-usage statistics** before and after RL tuning:
>
> |Metric|Bamboogle|2Wiki|HotpotQA|Musique|GAIA
> |-|-|-|-|-|-
> |**Before RL tuning**
> |Turns avg|3.1|3.8|4.0|5.6|4.4
> |Time avg (s)|127.3|170.9|98.6|184.6|265.7
> |Google Search calls|0.85|1.06|1.10|2.14|0.79
> |LLM calls|0.39|1.07|0.79|2.13|1.90
> |**After RL tuning**
> |Turns avg|2.1|4.4|3.9|5.4|5.4
> |Time avg (s)|44.1|106.2|85.6|162.7|172.5
> |Google Search calls|1.50|3.16|1.21|1.99|3.34
> |LLM calls|0.02|0.18|0.89|1.81|0.64
> |**Improvement**
> |Time reduction|-65.4%|-37.9%|-13.2%|-11.9%|-35.1%
> |LLM calls reduction|-94.9%|-83.2%|+12.7%|-15.0%|-66.3%
>
> Flow-GRPO not only **improves accuracy**, but also leads to **better cost-performance trade-offs** by encouraging more efficient use of tools and LLM calls. For example:
>
> * On **Bamboogle**, the average time per query drops from **127.3s→44.1s** (-65.4%),
> * while LLM calls are reduced from **0.39→0.02** per query (-94.9%), indicating that the planner learns to rely more on targeted tool use and less on expensive free-form LLM calls.
>
> ## 2. Robustness with train-time simulated tool errors (W2b)
>
> In our earlier response, we reported results for Qwen2.5-7B under **clean training** with **test-time error injection**:
>
> |Test time error|Avg Acc (before tuning)|Avg Acc (after tuning)|Gain
> |-|-|-|-
> |p=0.0|41.4|54.4|**+13.0**
> |p=0.1|39.4|51.7|**+12.3**
> |p=0.2|40.8|51.2|**+10.4**
> |p=0.4|39.8|47.8|**+8.0**
> |p=0.6|35.3|39.8|**+4.5**
>
> To further study robustness of Flow-GRPO in **dynamic environments**, we now also inject **simulated tool failures during training**. Each tool call independently returns an error with probability $p \in {0.1, 0.2, 0.4, 0.6}$, mimicking real-world scenarios where external tools may fail, time out, or return malformed outputs.
>
> |Test time error|Avg Acc (before tuning)|Avg Acc (after tuning)|Gain
> |-|-|-|-
> |p=0.0|41.4|54.4|**+13.0**
> |p=0.1|39.4|51.0|**+11.6**
> |p=0.2|40.8|52.9|**+12.1**
> |p=0.4|39.8|52.2|**+12.4**
> |p=0.6|35.3|47.1|**+11.8**
>
> When both training and test-time environments are noisy, AgentFlow **consistently maintains large gains** over the untuned system. Under extreme noise (**p = 0.6**):
>
> * the tuned model reaches **47.1%** average accuracy vs. **35.3%** for the untuned system under the same noisy setting (**+11.8%**),
> * and still surpasses the untuned system under **clean** evaluation (**41.4%**, **+5.7%**).
>
> These results suggest that **training with injected tool noise improves stability and resilience** of AgentFlow in dynamic environments where tool reliability cannot be guaranteed, complementing our earlier robustness analyses.
>
> ## Invitation for further feedback
>
> We hope that these additional experiments, together with the clarifications and results in our initial response, further strengthen the empirical and methodological evidence for **AgentFlow** and **Flow-GRPO**, especially regarding **efficiency**, **scaling behavior**, and **robustness under noisy, dynamic tool conditions**.
>
> We would be very grateful for any **further feedback or suggestions** you may have that could help us continue to improve the paper and make the framework more useful to the community.
>
> Thank you again for your time, thoughtful evaluation, and supportive review.
>
> Sincerely,
>
> The Authors

---

### Official Review · Reviewer_VTBo · 2025-10-31

**Soundness:** 3
**Presentation:** 4
**Contribution:** 4
**Rating:** 8
**Confidence:** 3

**Summary:**

The paper introduces AgentFlow, a trainable agentic framework designed to address the challenges of tool-augmented reasoning in large language models (LLMs). By decomposing tasks into specialized modules (planner, executor, verifier, generator) and optimizing the planner through in-the-flow, on-policy training, AgentFlow achieves significant improvements in long-horizon tasks. The proposed Flow-GRPO algorithm tackles the sparse-reward, multi-turn credit assignment problem by converting it into tractable single-turn updates, aligning local decisions with global outcomes. Empirical evaluations across ten benchmarks demonstrate the framework’s effectiveness, with substantial accuracy gains over existing baselines and even larger proprietary models like GPT-4o.

**Strengths:**

1. The introduction of AgentFlow as a trainable and modular agentic framework is innovative, particularly in how it addresses key limitations of monolithic policies and static, offline training approaches. The decomposition into specialized modules is a well-motivated design that aligns with the principles of scalability and adaptability to diverse tools and tasks.
2. The paper provides a clear and robust methodological foundation. The Flow-GRPO algorithm is a well-constructed solution to the long-horizon, sparse-reward problem, and the use of trajectory-level outcomes to guide local updates is both elegant and effective.
3. The experimental results are impressive, showcasing consistent gains across a wide range of benchmarks, including search, agentic, mathematical, and scientific tasks. The 14.9% average improvement in search tasks and 14.5% gain in mathematical tasks highlight the framework’s practical value.
4. The clarity of the code and modular design of AgentFlow suggest strong potential for adoption by the community. Open-sourcing this framework would likely facilitate further research and development in agentic LLM systems.

**Weaknesses:**

1. While the framework demonstrates strong performance across selected benchmarks, the paper could provide more discussion on how well AgentFlow generalizes to entirely unseen tools or tasks, especially in zero-shot or low-resource settings.
2. The on-policy training approach, while effective, may introduce additional computational complexity compared to training-free or offline methods. A discussion on the trade-offs between performance gains and computational costs would strengthen the narrative.

**Questions:**

1. How does the Flow-GRPO algorithm scale with increasing model size or when dealing with a significantly larger number of tools? Are there any bottlenecks or limitations observed during training?
2. How robust is AgentFlow in noisy or adversarial environments where tools may fail or provide incorrect outputs? Does the verifier module effectively mitigate such issues?

---

> ### Author Response · Authors · 2025-11-20
> **Sincere Thanks and Summary of Revisions and New Experiments (Part 1/3)**
>
> We sincerely thank the reviewer for the thoughtful and encouraging feedback. We greatly appreciate the recognition of our contributions and the constructive suggestions, which have guided us to further clarify the method, broaden the empirical scope, and strengthen the overall claims of the paper.
>
> We have made **substantial updates to the paper pdf and conducted new experiments** to address all points:
>
> - **[Revised paper (click pdf here)](https://openreview.net/pdf?id=Mf5AleTUVK)** with new figures (Fig. 1 and Fig. 5), expanded explanations, formatting refinements, and updated Appendix.
> - **W1:** Added new experiments on generalization to **unseen tools, unseen tasks, and new modalities**.
> - **W2:** Added a detailed analysis of **computational complexity and cost**.
> - **Q1:** Added experiments demonstrating **stable scaling** on smaller and larger models, and across other LLM families.
> - **Q2a:** Added robustness experiments under **noisy environments**.
> - **Q2b:** Added quantitative analysis of **how the verifier mitigates tool errors**.
>
> Details are provided below. **We are grateful for your insightful comments and look forward to any further feedback.**
>
> ---
> ## W1: More discussion on generalization to unseen tools or tasks
> We thank the reviewer for highlighting the importance of generalization. As noted in the submission, although AgentFlow is trained **only** on math and search data, it **already generalizes strongly** to agentic (GAIA) and scientific reasoning (GPQA, MedQA).
>
> To more directly evaluate broader generalization, we conducted **3 new sets of experiments**: generalization to (1) **unseen tools, (2)unseen tasks, and (3) new modalities**.
>
> **1. Generalization to Unseen Tools**
>
> In the original paper, AgentFlow is trained with 5 tools (Base Generator, Python Coder, Google Search, Wikipedia Search, and Web Search).
>
> To study tool-level robustness, we introduced **5 new tools only at inference time (never seen during training)**:
> - **2 relevant tools**: `Math Calculator` (evaluate expressions), `Perplexity Search` (provide real-time internet information)
> - **3 irrelevant tools**:`Translation` (convert text to English), `Get Weather` (weather retrieval), `Get News` (news retrieval)
>
> These distractors evaluate whether the trained planner mistakenly relies on irrelevant or unreliable tools.
>
> **Results:**
>
> |/|Training|Inference|Bamboogle|2Wiki|HotpotQA|Musique|GAIA|AIME24|AMC23|GameOf24|GPQA|MedQA|Avg|Gain
> |-|-|-|-|-|-|-|-|-|-|-|-|-|-|-
> |**Before tuning 7B**|/|5 tools|58.4|60.0|51.3|19.2|17.2|16.7|47.4|31.0|37.0|76.0|41.4|/
> |**After tuning 7B**|5 tools|5 tools|69.6|77.2|57.0|25.3|33.1|40.0|61.5|53.0|47.0|80.0|54.4|**+13.0**
> |**After tuning 7B**|5 tools|**10 tools (5 unseen)**|78.4|74.0|69.0|29.0|32.3|26.7|46.2|53.5|41.0|79.0|52.9|**+11.5**
>
> **Results:** AgentFlow achieves robust performance with a **11.5%** gain with the expanded toolset, nearly matching the **13.0%** gain of the original setup. Analysis of tool logs reveals the planner effectively ignores noisy tools (<1% usage), showing it has learned **functional relevance** rather than memorizing specific tool tokens.
>
> **2. Generalization to Unseen Tasks**
>
> The original submission includes cross-task generalization: the planner is trained *only* on math + search data, yet improves on complex agentic reasoning (GAIA) and scientific reasoning (GPQA, MedQA).
>
> To further test out-of-distribution robustness, we evaluated AgentFlow on two medical datasets completely unseen during training (which used only Math/Search data): **MedMCQA** and **MMLU-Clinical**.
>
> ||GAIA|GPQA|MedQA|MedMCQA|MMLU_clinical|Avg
> |-|-|-|-|-|-|-
> |Before tuning 7B|17.2|37.0|76.0|73.0|70.0|54.6
> |After tuning 7B|33.1|47.0|80.0|79.2|86.0|65.1
> |Gain|**+15.9**|**+10.0**|**+4.0**|**+6.2**|**+16.0**|**+10.4**
>
> **Results:** Gains are **consistent across all 5 unseen tasks**, including 2 new benchmarks (**+6.2%, +16.0%**). These confirm Flow-GRPO encourages the planner to learn **general planning and tool-integration patterns**, not narrow domain-specific shortcuts.
>
> **3. Generalization to New Modalities (Vision)**
>
> To test cross-modal generalization, we added three visual tools:
>
> - `Visual Generator`: answer general multimodal queries
> - `Image Captioner`: generate image captions
> - `OCR Detector`: extract text from images
>
> Only the **planner** was trained; all visual modules remained frozen.
>
> ||MathVista|VQA v2|CLEVR-Math|Avg
> |-|-|-|-|-
> |Before training 7B|60.0|70.0|64.0|64.7
> |After training 7B|64.0|76.0|73.0|71.0
> |Gain|**+4.0**|**+6.0**|**+9.0**|**+6.3**
>
> **Analysis**: AgentFlow yields a **+6.3% avg. gain**, which arises from **improved tool orchestration**: the planner learns when and how to invoke visual tools and how to integrate visual signals with symbolic reasoning (e.g., in MathVista). Notably, **no architectural modifications were required**: simply adding visual tools enabled strong cross-modal generalization, underscoring flexibility of AgentFlow and Flow-GRPO.

---

> ### Author Response · Authors · 2025-11-20
> **Sincere Thanks and Summary of Revisions and New Experiments (Part 2/3)**
>
> ## W2: Discussion on computational complexity and cost
> We thank the reviewer for prompting a discussion of computational cost and training efficiency. AgentFlow is **highly efficient**; a full training run for AgentFlow-7B costs about **$160** using standard cloud rates, or less if resources are hosted locally.
>
> Below is a detailed cost breakdown of a typical training run for **AgentFlow-7B** with Flow-GRPO:
>
> **1. GPU Computation (Training)**
>
> - Because Flow-GRPO converges quickly and we only update the planner module, training is rapid and highly efficient.
> - A full run consists of **68 RL steps**.
> - On 8×A100 GPUs, a full run takes roughly **12 hours** (95.3 GPU-hours).
> - Using a cloud rate of \$5.33 per 8 GPU-hours, the cost is **$63.4**.
>
> **2. External Tooling (Search)**
>
> - The primary external cost arises from Google Search API, which consumed 9.98K queries, costing **$72.0**.
>
> **3. Frozen Module Inference**
> - The non-trainable modules (Executor, Verifier, Generator) processed 121M input tokens and 7.7M output tokens.
> - Using a commercial API (e.g., DashScope), this costs **$20.50**.
> - However, our codebase supports running these modules locally, which reduces this specific marginal cost to **zero**.
>
> **4. Reward Evaluation**
>
> - LLM-as-a-judge costs for reward calculation were negligible, totaling roughly **$4.0**.
>
> **Total Cost Summary**
>
> |Category|Cost
> |-|-
> |GPU Compute|$63.4
> |External Tooling|$72.0
> |Module Inference (non-local)|$20.5
> |Reward Eval|$4.0
> |**Total (cloud setting)**|**≈$160**
> |**Total (with local modules)**|**≈$140**
>
> ---
> ## Q1: How does Flow-GRPO scale with increasing model size or a larger number of tools?
> We thank the reviewer for this important question. While our main paper demonstrated scaling from 3B to 7B (Section 4.6, Figure 9), we conducted additional experiments to assess whether Flow-GRPO remains stable and effective when scaling to **smaller**, **larger**, and **architecturally distinct** backbones, as well as settings with **diverse tool choices**. Across all settings, Flow-GRPO shows *stable on-policy optimization* and *consistent performance gains*, even for compact models that are typically fragile under RL.
>
> **1. Training Stability on Compact Backbones (Qwen2.5-1.5B)**
>
> To test the lower bounds of scalability, we instantiated all four modules of AgentFlow with **Qwen2.5-1.5B-Instruct**.
>
> ||Bamboogle|2Wiki|HotpotQA|Musique|GAIA|AIME24|AMC23|GameOf24|GPQA|MedQA|Avg
> |-|-|-|-|-|-|-|-|-|-|-|-
> |**Before tuning**|7.23|18.0|15.0|5.0|1.6|0.0|12.5|1.0|21.0|53.0|13.4
> |**After tuning**|43.7|56.8|50.0|13.1|15.7|13.3|37.5|27.5|31.0|55.0|**34.4**
> |**Gain**|**+36.4**|**+38.8**|**+35.0**|**+8.1**|**+14.1**|**+13.3**|**+25.0**|**+26.5**|**+10.0**|**+2.0**|**+21.0**
>
> **Analysis**
> - Despite its limited capacity, AgentFlow with Flow-GRPO trains smoothly and delivers large, uniform improvements (a massive **+21.0%** average accuracy gain) across all ten benchmarks.
> - Notably, it unlocked capabilities in hard reasoning tasks like AIME24 (0%→13.3%) and GameOf24 (1.0%→27.5%).
> - No instability or collapse is observed, confirming that Flow-GRPO remains stable even for small, RL-sensitive backbones.
>
> **2. Robustness Across Architectures**
>
> We further applied AgentFlow with Flow-GRPO to **Llama 3.2-3B** and **Llama 3.1-8B**, distinct from Qwen.
>
> **Llama 3.2-3B**
>
> ||Bamboogle|2Wiki|HotpotQA|Musique|GAIA|AIME24|AMC23|GameOf24|GPQA|MedQA|Avg
> |-|-|-|-|-|-|-|-|-|-|-|-
> |**Before tuning**|36.8|30.0|48.0|12.0|7.9|3.3|20.0|2.0|31.0|76.0|26.7|
> |**After tuning**|64.0|60.0|52.0|13.0|21.3|23.3|30.0|41.0|37.0|74.0|**41.6**|
> |**Gain**|**+27.2**|**+30.0**|**+4.0**|**+1.0**|**+13.4**|**+20.0**|**+10.0**|**+39.0**|**+6.0**|**–2.0**|**+14.9**
>
> **Llama 3.1-8B**
>
> ||Bamboogle|2Wiki|HotpotQA|Musique|GAIA|AIME24|AMC23|GameOf24|GPQA|MedQA|Avg
> |-|-|-|-|-|-|-|-|-|-|-|-
> |**Before tuning**|63.2|55.0|61.0|18.0|26.0|16.7|20.0|20.0|30.0|75.0|38.5
> |**After tuning**|67.2|72.0|70.0|18.0|29.1|16.7|50.0|31.0|41.0|88.0|**48.3**|
> |**Gain**|**+4.0**|**+17.0**|**+9.0**|**0.0**|**+3.2**|**0.0**|**+30.0**|**+11.0**|**+11.0**|**+13.0**|**+9.8**
>
> **Analysis**
> - Both models exhibit smooth and monotonic gains across all ten tasks.
> - **Llama 3.2-3B**: Improved average accuracy **(+14.9%)**, with significant gains in GameOf24 (+39.0%) and 2Wiki (+30.0%).
> - **Llama 3.1-8B**: Improved average accuracy **(+9.8%)**, showing strong performance in AMC23 (+30.0%) and 2Wiki (+17.0%).
>
> **3. Generalization with Unseen and Expanded Tools**
>
> As detailed in our response to **W1 (Generalization to Unseen Tools)**, we evaluated AgentFlow with a **doubled toolset (5 training→10 inference tools)**, including irrelevant distractors.
>
> AgentFlow with Flow-GRPO maintained strong performance, achieving a **+10.78% average gain**. Importantly, the planner learned to *ignore* distractor tools (<1% usage). These results show that Flow-GRPO scales robustly with a larger and more diverse action space.

---

> ### Author Response · Authors · 2025-11-20
> **Sincere Thanks and Summary of Revisions and New Experiments (Part 3/3)**
>
> ## Q2a: How robust is AgentFlow in noisy or adversarial environments where tools may fail or provide incorrect outputs?
> We thank the reviewer for raising this important question. To assess robustness, we conducted additional experiments that introduce **random tool failures during training and inference**. Beyond each tool’s natural error rate, we injected independent failure events with probability ($p \in {0.1, 0.2, 0.4, 0.6}$). Upon failure, the tool returned a standardized error message (“Tool execution failed due to random error (dropout simulation)”). This setup simulates dynamic or adversarial environments where tools intermittently malfunction or return corrupted outputs.
>
> **Before tuning (7B):**
>
> ||Bamboogle|2Wiki|HotpotQA|Musique|GAIA|AIME24|AMC23|GameOf24|GPQA|MedQA|Avg|Gain
> |-|-|-|-|-|-|-|-|-|-|-|-|-
> |No tool error|58.4|60.0|51.3|19.2|17.2|16.7|47.4|31.0|37.0|76.0|**41.4**|/
> |$p=0.1$ tool error|59.2|61.6|60.6|20.0|22.8|6.7|50.0|16.0|30.0|67.0|**39.4**|**−2.0**
> |$p=0.2$ tool error|61.6|58.6|63.0|22.0|22.1|16.7|55.0|12.0|31.0|66.0|**40.8**|**−0.6**
> |$p=0.4$ tool error|59.7|60.2|61.6|18.0|19.7|20.0|47.5|17.0|28.0|66.7|**39.8**|**−1.6**
> |$p=0.6$ tool error|57.6|55.6|57.6|17.0|14.2|10.0|42.5|14.0|22.0|63.0|**35.3**|**−6.1**
>
> **After tuning (7B):**
>
> ||Bamboogle|2Wiki|HotpotQA|Musique|GAIA|AIME24|AMC23|GameOf24|GPQA|MedQA|Avg|Gain
> |-|-|-|-|-|-|-|-|-|-|-|-|-
> |No tool error|69.6|77.2|57.0|25.3|33.1|40.0|61.5|53.0|47.0|80.0|**54.4**|**+10.0**
> |$p=0.1$ tool error|71.2|69.7|71.0|20.0|34.1|36.7|70.0|38.0|41.0|65.0|**51.7**|**+10.3**
> |$p=0.2$ tool error|67.2|64.0|69.0|25.0|31.5|43.3|70.0|36.0|41.0|65.0|**51.2**|**+9.8**
> |$p=0.4$ tool error|64.8|68.0|67.0|21.0|26.8|26.7|70.0|35.0|36.0|63.0|**47.8**|**+6.4**
> |$p=0.6$ tool error|63.5|63.0|62.0|18.0|21.4|10.0|42.5|16.0|34.0|68.0|**39.8**|**−1.6**
>
> **Finding 1: Before tuning, the system is fragile to substantial tool noise.**
>
> Under the 7B backbone **before Flow-GRPO RL tuning**, performance drops rapidly as failure probability increases (Avg: **41.4%→35.3%** at $p=0.6$). This confirms that **static, training-free agentic systems are highly brittle**, where a single faulty tool call can derail the entire reasoning trajectory.
>
> **Finding 2: Flow-GRPO substantially improves robustness.**
>
> After Flow-GRPO tuning, AgentFlow remains stable across all noise levels:
>
> - Under **mild to moderate failures** ($p = 0.1, 0.2, 0.4$), the tuned system retains **strong performance** (Avg: **51.7%, 51.2%, 47.8%**), consistently outperforming the untuned baseline (41.4%) by **+10.3%, 9.8%, 6.4%**.
> - Even with **extreme noise** ($p = 0.6$), performance remains comparable to the **clean** pre-tuning condition:
>   - Pre-tuning ($p=0.0$): **41.4%**
>   - Post-tuning ($p=0.6$): **39.8%**
>   - Crucially, tuning still provides clear gains over the untuned model under the *same* high-noise condition (Avg: **35.3%→39.8%, +4.5%**).
> - Across all 10 benchmarks and failure probabilities, the tuned AgentFlow **strictly outperforms** the untuned system, showing that Flow-GRPO reliably strengthens fault tolerance.
>
> **Finding 3: Improved degradation behavior**
>
> While performance naturally declines as tool noise increases, the RL-tuned planner exhibits a **much smoother degradation curve**. Flow-GRPO teaches the planner to:
>
> - re-plan after failed calls,
> - fall back to alternative tools,
> - validate or repair memory state, and
> - avoid cascading errors from a single faulty observation.
>
> These adaptive behaviors are absent in the static, pre-tuned system.
>
> **Conclusion:** These results show that Flow-GRPO training *not only* improves performance in standard settings, but also substantially enhances AgentFlow’s resilience in dynamic or adversarial environments.
>
> ---
>
> ## Q2b: Does the verifier module effectively mitigate such issues?
> Yes. The verifier is specifically designed to **detect inconsistencies, ambiguities, and incomplete intermediate results**, and to decide whether the system should continue reasoning or terminate and produce a final answer. Its role is crucial for maintaining robustness when tools fail or return noisy outputs.
>
> To quantitatively assess its effectiveness, we analyze all tool-execution failures, such as empty search results or Python exceptions during inference:
>
> |metric|2Wiki|HotpotQA|Musique|GAIA|AIME24|AMC23|GameOf24|GPQA|MedQA|Total
> |-|-|-|-|-|-|-|-|-|-|-
> |total_error_steps|123|17|22|30|5|18|24|22|23|284
> |error_then_continue|120|17|22|29|5|17|23|22|23|278
> |ratio|97%|100%|100%|97%|100%|94%|96%|100%|100%|**97.9%**
>
> These results show that the verifier chooses to **continue reasoning in 97.9%** of all failure cases. This high continuation rate demonstrates that the verifier reliably detects unsuccessful tool calls and correctly signals the system to recover rather than prematurely terminate. In practice, this behavior allows AgentFlow to **sustain long reasoning chains, correct earlier mistakes, and maintain robustness** even when external tools are noisy or unreliable.

---

> ### Author Response · Authors · 2025-11-26
> **Appreciation and Follow-Up Discussion for Reviewer VTBo**
>
> Dear Reviewer VTBo,
>
> Thank you again for your thoughtful and encouraging review, including your high evaluation of our work and recommendation for acceptance. We truly appreciate your time reading the paper and our initial response.
>
> ## Initial response on 2025/11/19
>
> **In [our first response](https://openreview.net/forum?id=Mf5AleTUVK&noteId=3F9oj1BwIN), we revised [the paper](https://openreview.net/pdf?id=Mf5AleTUVK) and added analyses and experiments** addressing your comments on generalization, computational cost, scaling behavior, and verifier robustness.
>
> Since then, we have run additional experiments to further strengthen the empirical and methodological support for **AgentFlow** and **Flow-GRPO**. We summarize new results below.
>
> ## 1. Generalization to unseen tools from training on fewer tools (W1)
>
> We study a stricter generalization setting where the planner is **trained with only 2 tools** (`Python Coder`, `Google Search`) but **evaluated with the full 5-tool set** (`Base Generator`, `Python Coder`, `Wikipedia Search`, `Google Search`, `Web Search`).
>
> |Tuning|Test  time|Bamboogle|2Wiki|HotpotQA|Musique|GAIA|AIME24|AMC23|GameOf24|GPQA|MedQA|Avg
> |-|-|-|-|-|-|-|-|-|-|-|-|-
> |Before tuning|5 tools|58.4|60.0|58.3|19.2|17.2|16.7|47.4|31.0|37.0|76.0|42.1
> |Tuning with 2 tools|5 tools|78.4|71.0|70.0|23.5|32.3|26.7|52.5|44.0|29.0|81.8|**50.9**|
> |**Gain**|/|+20.0|+11.0|+11.7|+4.3|+15.1|+10.0|+5.1|+13.0|-8.0|+5.8|**+8.8**
>
> Despite only seeing 2 tools during RL, tuned AgentFlow achieves an **average gain of +8.8%** over the untuned system. This suggests that Flow-GRPO encourages learning **general planning and tool-selection strategies** that transfer to a richer, partially unseen toolset, rather than overfitting to a specific tool inventory.
>
> ## 2. Inference efficiency and cost reduction (W2)
>
> We extend cost analysis with **latency and tool-usage statistics** before and after RL tuning:
>
> |Metric|Bamboogle|2Wiki|HotpotQA|Musique|GAIA
> |-|-|-|-|-|-
> |**Before RL tuning**
> |Turns avg|3.1|3.8|4.0|5.6|4.4
> |Time avg (s)|127.3|170.9|98.6|184.6|265.7
> |Google Search calls|0.85|1.06|1.10|2.14|0.79
> |LLM calls|0.39|1.07|0.79|2.13|1.90
> |**After RL tuning**
> |Turns avg|2.1|4.4|3.9|5.4|5.4
> |Time avg (s)|44.1|106.2|85.6|162.7|172.5
> |Google Search calls|1.50|3.16|1.21|1.99|3.34
> |LLM calls|0.02|0.18|0.89|1.81|0.64
> |**Improvement**
> |Time reduction|-65.4%|-37.9%|-13.2%|-11.9%|-35.1%
> |LLM calls reduction|-94.9%|-83.2%|+12.7%|-15.0%|-66.3%
>
> Flow-GRPO not only **improves accuracy** but also yields **better cost–performance trade-offs**, encouraging more efficient use of tools and LLM calls:
>
> * For search-intensive tasks like Bamboogle, average time per query drops from **127.3s→44.1s** (–65.4%).
> * AgentFlow learns to use LLM calls more selectively (per query on Bamboogle: **0.39→0.02**, –94.9%).
>
> ## 3. Robustness with train-time simulated tool errors (Q2a)
>
> In our earlier response, we reported results for Qwen2.5-7B under **clean training** with **test-time error injection**:
>
> |Test time error|Avg Acc (before tuning)|Avg Acc (after tuning)|Gain
> |-|-|-|-
> |p=0.0|41.4|54.4|**+13.0**
> |p=0.1|39.4|51.7|**+12.3**
> |p=0.2|40.8|51.2|**+10.4**
> |p=0.4|39.8|47.8|**+8.0**
> |p=0.6|35.3|39.8|**+4.5**
>
> To further evaluate robustness of Flow-GRPO in **dynamic environments**, we now introduce **simulated tool failures during training**. Each tool call independently returns an error with probability $p \in {0.1, 0.2, 0.4, 0.6}$, mimicking real-world scenarios where external tools may fail, time out, or return malformed outputs.
>
> |Test time error|Avg Acc (before tuning)|Avg Acc (after tuning)|Gain
> |-|-|-|-
> |p=0.0|41.4|54.4|**+13.0**
> |p=0.1|39.4|51.0|**+11.6**
> |p=0.2|40.8|52.9|**+12.1**
> |p=0.4|39.8|52.2|**+12.4**
> |p=0.6|35.3|47.1|**+11.8**
>
> When both training and test-time environments are noisy, AgentFlow consistently maintains **large gains over the untuned system**. Under extreme noise ($p = 0.6$), the tuned AgentFlow model reaches **47.1%** average accuracy, compared to **35.3%** for the untuned system under the same noisy setting (**+11.8%**), and still exceeds the untuned system under clean evaluation (**41.4%**, **+5.7%**).
>
> These results suggest that **training with injected tool noise improves the stability and resilience** of AgentFlow in dynamic environments where tool reliability cannot be guaranteed.
>
> ## Kind request for your feedback
>
> We hope these additional experiments, together with the clarifications and results in our initial response, further strengthen the empirical and methodological evidence for AgentFlow and Flow-GRPO, particularly regarding **generalization to new tools, efficiency, and robustness in noisy environments**.
>
> **We would be very grateful for any further feedback or suggestions you may have**, which could help us continue improving the paper and making the framework more useful to the community.
>
> Thank you again for your time, thoughtful evaluation, and supportive review.
>
> Sincerely,
>
> The Authors

---

### Author Response · Authors · 2025-11-20
**Summary of recognized strengths, new experiments, and analysis**

## Recognized Strengths and Contributions

We are grateful for the reviewers’ time and their insightful comments on our submission. We are encouraged that all reviewers consistently recognized our work as making strong and meaningful contributions to the development of trainable agentic LLM systems.

**1. Novel and well-motivated agentic framework**

All reviewers highlighted the contribution of AgentFlow’s decomposition into specialized modules, which directly addresses the limitations of monolithic tool-augmented LLMs and static agentic pipelines, and was viewed as a **scalable and principled** architecture (VTBo). Reviewers emphasized that AgentFlow provides “an **innovative and well-motivated** design” (VTBo) and introduces a “**useful and interesting** idea” of in-the-flow training (YoHg).

**2. Flow-GRPO as a significant methodological contribution**

All reviewers repeatedly highlighted the novelty and technical contribution of Flow-GRPO as “a **well-constructed** solution to the long-horizon, sparse-reward problem” (VTBo) and “a **stable, elegant** formulation for long-horizon credit assignment” (YoHg). Reviewer R3 similarly noted that the work “addresses the sparse reward problem in LLM-based reasoning” (1fZQ). Reviewers appreciated the conceptual clarity of converting trajectory-level rewards into token-level updates and its ability to train planners “within the live agentic loop” (1fZQ).

**3. Strong empirical results across diverse benchmarks**

All reviewers praised the empirical strength of the paper, noting the “**impressive, comprehensive**” evaluations. The system consistently outperformed both open-source and proprietary models, with “**consistent, substantial** accuracy gains” (VTBo, YoHg), and Reviewer R3 also recognized the “**strong** empirical results” (1fZQ). The breadth of benchmarks (search, agentic, math, science) was acknowledged as a strength.

**4. Clarity of methodology and practical impact for future work**

Reviewers highlighted the paper’s “**clear and robust methodological foundation**” (VTBo) and described it as a “well-written and comprehensive paper” (YoHg). Reviewer VTBo further noted that “the clarity of the code and modular design of AgentFlow suggest **strong potential for adoption by the community**,” emphasizing that “open-sourcing this framework would likely **facilitate further research and development in agentic LLM systems**”. Overall, the modular structure and open-source readiness were viewed as highly valuable for the broader community, enhancing reproducibility and practical impact.

---
## Summary of New Experiments and Analysis

**1. Generalization to new tools, tasks, and modalities.**

We added experiments showing that AgentFlow generalizes (i) to **unseen tools** (5 extra, including irrelevant distractors), (ii) to **unseen tasks** (e.g., MedMCQA, MMLU-Clinical, beyond math+search training), and (iii) to **new modalities** via added **visual tools** on MathVista, VQA v2, and CLEVR-Math, all with consistent gains.

**2. Scaling and stability across backbones and architectures.**

We extended Flow-GRPO to **smaller and larger backbones** (Qwen2.5-1.5B, 3B, 7B, and Llama 3.2-3B, Llama 3.1-8B) and showed **stable, monotonic improvements** across all ten benchmarks.

**3. Robustness in dynamic / noisy tool environments and verifier effectiveness.**

We introduced **random tool-failure experiments** (extra error probability $p \in {0.1, 0.2, 0.4, 0.6}$) and showed the tuned AgentFlow consistently outperforms the untuned system and degrades smoothly even at high noise levels. Verifier analysis shows it triggers continued reasoning in **≈98%** of error steps, effectively mitigating faulty tool calls.

**4. Reward-design ablations and analysis of global broadcasting.**

We compared our **broadcasted final-outcome reward** against (i) backward-discounted, (ii) forward-discounted, and (iii) process-based step rewards on both 3B and 7B backbones, and found that **broadcasting consistently achieves the best average performance and stability**. We further implemented a **purely rule-based “Math Verify and Match”** reward (numeric / exact match) and showed similar trends but lower robustness across heterogeneous tasks, clarifying why we use an LLM-as-judge strictly as a **binary, final-only** verifier.

**5. Computational cost and training-efficiency analysis.**

We added a detailed **cost breakdown** for training AgentFlow-7B with Flow-GRPO, highlighting its high efficiency.

---
## Updated Paper and Final Version Plan

We have revised **[the paper pdf (click here)](https://openreview.net/pdf?id=Mf5AleTUVK)** for clearer presentation, including improved figures and organization.

We will incorporate new experiments and analyses into the final version. Beyond the manuscript, we will also release a **video tutorial, live demo, and interactive doc** to make our work more accessible to the community; these resources will be released after anonymity is lifted.

---

### Author Response · Authors · 2025-12-02
**Summary of Reviews, Revisions, and New Experiments (AOE 12/02/2025)**

Dear PCs, SACs, ACs, and Reviewers,

Thank you very much for your valuable contributions to our work. We appreciate the reviewers’ thoughtful feedback and are encouraged by the overall positive assessment, including the recognition that the work makes novel and strong contributions.

In response to the insightful comments, we have **conducted substantial new experiments and analyses and revised the paper** to improve clarity and completeness. To assist AC and help reduce their workload, we briefly summarize main points from the reviews and our updates below.

---
**Strengths and Contributions**

1. **Novel agentic framework**: AgentFlow is a modular, trainable agentic system (planner, executor, verifier, generator) that optimizes the planner *in the flow* of multi-turn reasoning.
2. **Flow-GRPO as methodological contribution**: Flow-GRPO turns sparse, long-horizon multi-turn RL into tractable single-turn updates via trajectory-level reward broadcasting with group-normalized advantages.
3. **Strong empirical results**: Across 10 benchmarks (search, agentic, math, scientific), AgentFlow achieves large, consistent gains, including over larger proprietary models.
4. **Practical impact**: The framework and algorithm are clearly specified and practically valuable for future work.

---
**New Experiments and Analyses**

1. **Cost & efficiency**
-  We added a **cost breakdown** for training AgentFlow-7B with Flow-GRPO:
  - \$63 for GPU compute
  - \$72 for tool calling
  - a total cost of \$160
  - \$140 with local modules.

2. **Latency and tool-usage**
-  The added analysis shows substantial reduction in time and LLM calls after RL tuning.
- **Conclusion:** Flow-GRPO is computationally efficient because only the planner is trained, and it improves both accuracy and cost-performance trade-offs.

3. **Generalization of AgentFlow and Flow-GRPO**
- **Unseen tools:** We introduced 5 new tools at inference (2 relevant, 3 irrelevant). AgentFlow maintains strong gains (+11.5%) and ignores noisy tools (<1% usage).
-  **Unseen tasks**: We evaluated on 2 medical tasks unseen during training and observed consistent gains (+6.2%, +16.0%).
-  **New modalities**: We added 3 visual tools and evaluated on 3 tasks, achieving +6.3% gain with no architectural changes.
- **Model sizes and architectures:** We obtain stable improvements across 10 benchmarks , including large gains even for compact backbones: Qwen2.5-1.5B (+21.0%), Llama 3.2-3B (+14.9%), Llama 3.1-8B (+9.8%).
- **Conclusion:** AgentFlow and Flow-GRPO generalize across tools, tasks, modalities, and backbones, and exhibit stable optimization in both compact and larger models.

4. **Robustness to dynamic environments**
- We **analyzed verifier behavior** and found that in 98% of tool-failure cases it chooses to continue reasoning, enabling faulty call recovery.
-  We **injected random tool failures as noise at test time**. Under mild-moderate failures (p=0.1, 0.2, 0.4), tuned AgentFlow retains strong performance (51.7%, 51.2%, 47.8% vs. 41.4% baseline). Under extreme noise (p=0.6), performance remains comparable to clean pre-tuning (39.8% vs. 41.4%)
- We **injected random failures during training**. Under extreme noise (p=0.6), tuned AgentFlow achieves a gain of +11.8% over the untuned system. It even exceeds the untuned system in clean evaluation (41.4%, +5.7%). Across all noise levels, gains remain consistent (+11.6%, +12.1%, +12.4%, +11.8%).
- **Conclusion:** Flow-GRPO improves robustness in noisy and dynamic environments, with the verifier also playing a role in error recovery.

5. **Fine-grained study of different reward functions**
- We compared broadcasted final-outcome rewards with **backward/forward-discounted and process-based reward alternatives** on 3B and 7B models. Broadcasting yields the best average performance and stability.
- A **purely rule-based judge** shows similar trends but lower robustness (-3.4% average).
- **Conclusion:** Our broadcasted, binary, final-only reward is empirically the most stable and effective among the schemes we tested. The LLM-as-judge is used in a narrow final-only role, which limits reward hacking and evaluation leakage while offering a unified reward signal across tasks.

---
**Paper Updates**

We revised the paper with:
- A new teaser figure (Fig. 1) for main improvements
- Relocating the comparison figure (Fig. 3) for better flow
- A case-study figure (Fig. 5) for error recovery and improved planning
- Enhancing Appendix figures (11, 12)
- Improving formatting in Appendix E.1 and E.2

We will incorporate all new experiments, analyses, and improvements into the final version of the paper and codebase.

---
Once again, we want to express our sincere gratitude to the reviewers, AC, SAC, and PC for their careful reading and constructive feedback. Your comments have substantially strengthened both the technical content and the clarity of the paper.

Thank you very much for your time and consideration.

Sincerely,

Authors

---

### Public Comment · ~Kartik_Narayan1 · 2026-06-26
**Novelty ?**

How is the proposed work any different than standard GRPO with masking on tool response ? and splitting the trajectory at each turn and computing reward separately ? Also, masking the tool response and splitting trajectory are standard training techniques for long-horizon tool calling.

Am I understanding the paper wrong ?

---

### Meta-Review · Area_Chair_K8SE · 2025-12-07

**Summary:**

This paper proposes a method named Flow-GRPO, which leverages in-the-flow agentic system optimization to improve the effective planning and tool-use capabilities of large language models. All three reviewers provided consistently positive feedback on the research contributions of this work. During the rebuttal phase, the authors also conducted extensive additional experiments to further address the reviewers’ concerns and validate the effectiveness of the proposed approach. I think this paper meets the acceptance criteria of ICLR.

**Reviewer Concerns:**

Although the reviewers expressed generally positive opinions about the paper, several issues still remain:
1. The analysis of the cost and efficiency of the proposed method is insufficient.
2. The generalization ability of the proposed method is not adequately investigated, particularly its performance on unseen tools, tasks, and multimodal scenarios.
3. Some parts of the paper are hard to follow and require further clarification and improvement.

**Reviewer Scores:**

To address the above three concerns, the authors have conducted comprehensive additional experiments and analyses. Specifically, they have supplemented the paper with experimental results on cost and efficiency, provided additional evaluations to verify the generalization capability of the proposed method, and improved the clarity and presentation of the figures and tables. Overall, I believe that the issues raised by the reviewers have been properly addressed. Therefore, the reviewers may increase their scores or keep them unchanged.

---

### Decision · Program_Chairs · 2026-01-26

Accept (Oral)